# Predictive values of inflammatory back pain, positive HLA B27 antigen and acute and chronic magnetic resonance changes in early diagnosis of Spondyloarthritis. A study of 133 patients

Liliya Yankova Komsalova[1]*, Mª Pilar Martínez Salinas[2], José Fermín Gómez Jiménez[2]

1 Rheumatology Department, Hospital Marina Salud, Denia, Alicante, España, 2 Radiology Department, Hospital Marina Salud, Denia, Alicante, España

* liliya.yankova.kom@gmail.com

## Abstract

### Objectives

To analyse the predictive values of inflammatory back pain (IBP), positive HLA B27 antigen, increased C-reactive protein (CRP), Spondyloarthritis (SpA) features, familial history (FH), magnetic resonance sacroiliac joints (MRI-SIJ) imaging and its weight in early SpA diagnosis.

### Methods

133 patients with back pain, aged <50, duration of the pain <2 years were included. Data such as IBP, HLA B27, increased CRP, SpA features, FH, SIJ´s radiography and MRI were collected for each patient. STIR sequences were classified as strongly positive bone morrow oedema (SPBME ≥2), clearly present and easily recognisable as positive according to the ASAS criterion, weakly positive (WPBME ≥2), suggestive, but not easily recognisable and, clearly negative none of those features. T1-weighted sequences were assessed as positive/negative for erosion, fat metaplasia, backfill and sclerosis, if ≥1, for each lesion was present. MRI images were read by three blinded readers.

### Results

The average age was 38.9 years. 47 (35.3%) patients received SpA diagnosis according to the clinical opinion. IBP was highly specific, 0.81 and sensitive, 0.83. HLA B27 was positive in a half of the SpA patients. SPBME ≥2 provided a great specificity, 0.94 and an acceptable sensitivity, 0.79. Erosion was significantly more frequent in SpA patients (72% vs 7%), specificity 0.93. The addition of erosion ≥1 to the WPBME ≥2 noticeably improved specificity, 0.98, although slightly decreased sensitivity, 0.64. Fat metaplasia and backfill were highly specific, but poorly sensitive. Factors forecasting positive diagnosis were IBP, followed by SpA features and increased CRP.

**Data Availability Statement:** All relevant data are within the manuscript and its Supporting Information files.

**Funding:** The authors received no specific funding for this work.

**Competing interests:** The authors have declared that no competing interests exist.

## Conclusions

At the onset, IBP might be a good marker for selecting patients with suspicion of SpA. The addition of erosion to the ASAS criterion might be helpful for early diagnosis, especially in patients with doubtful STIR imaging where BME is present but it is hard to determinate whether the ASAS "highly suggestive" criterion is met.

## Introduction

SpA are a wide group of inflammatory rheumatic diseases with distinct clinical and genetic characteristics. According to their clinical presentation, patients have predominantly axial or peripheral SpA. Aetiology remains unknown. A major genetic predisposition was suggested in the ´70s and confirmed latterly by discovering the Human leucocyte antigen B27 as responsible for the disease susceptibility [1]. Early diagnosis of SpA has long been the major challenge in daily practice. The first approach to the diagnosis was the New York criterion based on radiographic sacroiliitis and showed a good specificity, nonetheless sensitivity was poor [2]. Later years, Amor and ESSG criteria introduced SpA clinical features in an attempt to improve sensitivity. The diagnostic accuracy of both was compared in several studies, confirming a better performance in patients with definite diagnosis, and a higher classification rate for ESSG, followed by Amor and Berlin criteria [3,4]. Thereafter, MRI-SIJ imaging together with 2 or 3 SpA features was found to be able to increase likelihood of positive diagnosis to 90% in patients without radiographic sacroiliitis [5]. New evidence based on TNF- alfa blockers use guided us to the idea that absence of radiographic sacroiliitis did not exclude diagnosis and those patients could be considered as patients with pre-radiographic phase of SpA [6]. The new data led to the current proposal of classification criteria by The Assessment of Spondyloarthritis International Society (ASAS) group for patients without definitive radiographic sacroiliitis [7]. The criteria was based in two arms, "imaging arm", patients with positive sacroiliitis on imaging (radiograph or MRI) plus at least one SpA feature, and "clinical arm", HLA B27 positive patients who had at least two SpA features [8,9]. Definition of positive MRI-SIJ imaging was developed by consensus in 2009, updated in 2016, and required evidence of subchondral bone marrow oedema depicted as hyperintense signal on short tau inversion recovery (STIR) or T2 fat suppression (FS) sequences, clearly present and located in a typical anatomical area, highly suggestive of SpA. BME meeting the above criterion had to be seen on at least two consecutive slices of an MRI scan. Detection of inflammation on a single slice might be sufficient if more than one inflammatory lesion was present. In cases where inflammatory lesions were present but it was hard to determine whether they met the" highly suggestive" criterion, T1-weighted chronic lesions (Tw1CL), especially erosion, were defined as contributing to the diagnosis, suggestive, but not determinants [10,11]. Unfortunately, BME was shown to be easily detected in sacroiliac joints of healthy individuals, those with known mechanical strain acting upon, and women with postpartum back pain [12]. At the same time Tw1CL were observed even in the absence of MRI inflammation in patients with non-radiographic axial SpA (nr-axSpA), and thus raising the question of how reliable these various lesions were, and what role did they play in the early diagnosis of SpA [13]. In this study we try to answer two questions; First, to evaluate the predictive validity of the different SpA items and determine whether structural changes, mainly erosion, could contribute to early SpA diagnosis, and Second, to find factors forecasting positive diagnosis.

## Patients and methods

### Clinical evaluation

We prospectively included 133 consecutive patients with low back pain (LBP), aged <50 years, duration of the pain <2 years, referred to our department between 2014 and 2018. Data such as IBP, HLA B27 antigen, increased CRP, FH, and SpA features, such as arthritis, dactylitis, psoriasis, uveitis, inflammatory bowel diseases and enthesitis were collected for each patient. SIJ´s radiography and MRI were performed on each patient.

All patients were evaluated during the follow-up period by the rheumatologist and final diagnosis was based on his clinical opinion, settled on clinical examination, laboratory values and imaging (MRI and radiographies). Where possible, Modified New- York criterion was applied. The authors realized that in patients with pre-radiographic sacroiliitis the diagnosis was influenced by the ASAS group classification criteria, nonetheless the clinical opinion was mandatory for the final diagnosis and this used as a gold standard for the study purpose. Patients who ended up with positive diagnosis were classified as having non radiographic SpA (nr-axSpA) or Ankylosing spondylitis (AS). The LBP was defined as IBP or non-specific back pain (NSBP), according to the Berlin, Calvin or ASAS criterion. Patients had to fulfil at least 2/4 parameters for the Berlin, and 4/5, for the Calvin or ASAS criterion [14,15]. CRP (normal range 2–10 mg/l) was measured and HLA B27 typing was performed.

### Ethics statement

The study protocol was approved by the hospital´s Clinical Research Committee. Authors were guided by The Helsinki Declaration of 1975, as revised in 2000 for good clinical practice. Patients were included as a consecutive number as entering on the study protocol. Confidentiality was strictly maintained by anonymising patients and erasing any personal date. Readers were blinded to personal and clinical data, time order and diagnosis in any of the imaging modality. An anonymised database was used for statistical analysis.

### MRI evaluation

Scans included SIJ´s coronal Tw1 spin-echo, axial Tw1 turbo spin-echo, coronal T2-weighted (Tw2) turbo spin-echo, axial and coronal oblique STIR sequences. All scan parameters are exposed in S1 Table. Assessment was based on Tw1 and STIR sequences. All Tw1 and STIR images were blinded for time order, diagnoses, and clinical data. The STIR images were evaluated by the rheumatologist and the first radiologist. Discordant cases were contrasted by a third reader (2nd radiologist), regarded as conclusive, reaching 100% of agreement.

To define active sacroiliitis on STIR sequences we used the ASAS/ OMERACT consensual approach for defining active sacroiliitis and the ASAS working group update, accepting for positive MRI the presence of a strong hyperintense signal located periarticularly on at least two consecutive slices of an MRI scan. If there was BME in only one slice, then more than one inflammatory lesion was required [11,13]. For simplicity we used the abbreviation BME ≥2. Moreover, we divided STIR sequences into three categories, strongly positive (SPBME ≥2), if images fulfilled ASAS/OMERACT criterion and they were clearly present and easily recognizable as positive by readers (high confidence level), and weakly positive (WPBME ≥2), if lesions fulfilled the above criterion, but they were tenuous and not easily identifiable as positive (low confidence level). Patients without any of those lesions were qualified as having clearly negative (CNBME ≥2) STIR imaging. The use of a semiquantitative way instead of a quantitative scoring for defining STIR imaging was an attempt to better reflect a circumstance mentioned by the ASAS guidelines for application of a definition of a positive MRI, where BME appeared

to be present but it was hard to determine if the "highly suggestive" criterion was met [11]. We believe that is a situation frequently faced by clinicians in their daily practise.

To define Tw1 chronic inflammatory lesions we used the standardised definition developed by Canada-Denmark MRI working group for erosion, fat metaplasia, backfill and sclerosis. Erosion was defined as full thickness loss of the dark appearance of either iliac or sacral cortical bone at its anticipated location, and loss of the normal bright appearance of the adjacent bone marrow. Fat metaplasia was defined as an increased signal in bone marrow on T1 sequences, and backfill as the complete loss of iliac or sacral cortical bone at its anticipated location and increased signal, clearly demarcated from the adjacent normal marrow by an irregular dark signal, reflecting sclerosis at the border of the eroded bone [16]. Tw1CL were assessed separately as a binary variable (positive/negative), by each reader (the rheumatologist and the two radiologists) for erosion, fat metaplasia, backfill and sclerosis accepting for positive image the presence of at least one of those lesions ($\geq 1$) in each category, and for negative, if none of those lesions were present. Concordance between any two readers was accepted as conclusive. Readers were previously calibrated with a training session and prereading exercises were developed, setting the threshold for detection of the acute and chronic MRI lesions.

## Radiological evolution

Sacroiliac joint radiographies were evaluated by the rheumatologist and assessed according to The Modified New York criteria [2]. Radiographies were graded as positive if there were at least bilateral grade 2, or unilateral grade 3 sacroiliitis and as negative if none of those features were present.

## Statistical analysis

SPSS Version 25 and Epidata 4.2 were used for data analysis. All end-point variables were tested against the gold standard (i.e. the rheumatologist's diagnosis at the end of the follow-up period). For descriptive purposes, the results were shown as relative frequencies (percentages) for categorical variables, and as the mean (SD) for continuous variables. T student, Mann-Whitney test, and Chi-square test (or Fisher's exact test) were employed to compare variables between groups. Diagnostic utility, measured by sensitivity, specificity, and positive and negative predictive values (PPV, NPV) of each SpA item was calculated. Statistical significance was set at p <0.05. Multivariate logistic regression binary analysis was used to determine factors forecasting positive diagnosis. Associations were expressed using Odds ratios (ORs) with 95% confidence interval (95% CI). Kappa concordance coefficient was used to assess the interobserver agreement.

## Results

### Patients characteristics

The average age in our study was 38.9 years with male/female ratio 0.41/1. 47 (35.3%) patients received SpA diagnosis according to the clinical opinion. 8 (17%) patients had radiographic sacroiliitis. IBP was found in 41.4% of patients, more often in patients with positive SpA diagnosis, 83% vs 18.6% SpA/not SpA patients. Positive HLA B 27 antigen in 30.3% of them, 49% vs 20% SpA/not SpA; increased CRP in 18.8%, 40.4% vs 7% SpA/ not SpA; and SpA features in 18.8%, 36.2% vs 9.3% SpA/not SpA patients. The most common SpA features were arthritis and/or dactylitis, 9 patients, followed by psoriasis 7, uveitis 5, inflammatory bowel diseases 3, and enthesitis, 1 patient. Only 14 (10.5%) patients reported FH. No differences were found between gender and positive SpA diagnosis (p = 0.093). All patients' characteristics are exposed in S2 Table.

## Predictive values

Patients with positive SpA diagnosis had more often IBP, sensitivity 0.83, specificity 0.81 (p < 0.001). HLA B27 was positive in a half of the SpA patients, sensitivity 0.49, specificity, 0.80 (p <0.001). CRP was more frequently increased in SpA patients, sensitivity 0.40, specificity 0.93 (p <0.001). Patients with SpA had more often SpA features, specificity 0.91, nonetheless sensitivity was lower 0.36 (p <0.001) (see S3 Table). We did not find statistical differences between patients with and without positive FH (p = 0.225).

## MRI findings

SPBME $\geq$2 was detected in 31.6% of the patients, WPBME $\geq$2 in 21.1% and CNBME $\geq$2 in 47.4% of them. SPBME $\geq$2 was significantly more often detected in patients with positive SpA diagnosis according to the clinical opinion (78.7% vs 5.8%), showing a high sensitivity, 0.79, and an excellent specificity, 0.94 (p<0.001). WPBME $\geq$2 was reported in 87% vs 33.7% of the SpA/ not SpA patients, bringing a good sensitivity 0.87, nonetheless specificity was slightly lower, 0.66 (p <0.001). Erosion $\geq$1 was significantly more often found in patients with positive SpA diagnosis (72% vs 7%) (p <0.001) and it was sensitive, 0.72 and strongly specific 0.93, providing an excellent PPV 0.85 (p <0.001). Fat metaplasia, sensitivity 0.57, specificity, 0.83 and sclerosis, sensitivity 0.62, specificity 0.80 were also more frequently seen in the SpA group (p <0.001). Backfill was found to be highly specific, 1.0, unfortunately sensitivity was excessively poor, 0.39 (p<0.001). The association of SPBME $\geq$2 with erosion $\geq$1 significantly improved specificity, 1.0, although, mildly decreased sensitivity, 0.62 (p <0.001). The association of WPBME$\geq$ 2 with erosion $\geq$1 was highly specific, 0.98, but moderately sensitive 0.64 (p <0.001). The addition of erosion $\geq$1 and sclerosis $\geq$1 to the SPBME $\geq$ 2, as well to the WPBME $\geq$2 noticeably increased specificity (1.0; 0.98, respectively); nonetheless, sensitivity remained poor (0.51; 0.51 respectively). The addition of erosion $\geq$ 1 and backfill $\geq$1 to the SPBME $\geq$2, as well as to the WPBME $\geq$2 markedly increased specificity (1.0; 1.0 respectively); however, scarcely contributed to early diagnosis, sensitivity (0.28, 0.28 respectively) (p<0.001). All MRI outcomes are shown in S3 Table.

## Multivariate analysis

S4 Table shows a multivariate binary analysis, searching for factors forecasting a positive diagnosis. Analysis was performed for the entire group of SpA patients (AS and nr-axSpA). Selected variables were IBP, HLA B27, CRP, SpA features, positive sacroiliitis on radiography, SPBME $\geq$2, WPBME $\geq$2, erosion, fat metaplasia, backfill, sclerosis $\geq$1, and the different associations of the SP/WPBME $\geq$2 with the Tw1CL. Variables such as positive radiography and MRI (Tw1 and STIR sequences), were found to lead to collinearity problems and biased coefficient estimation, therefore, excluded from the data analysis. IBP, increased CRP, and SpA features were uncorrelated variables, hence, evaluated as variables forecasting positive diagnosis. Associations were expressed using Odds Ratio (OR). The variables significantly related to positive SpA diagnosis were IBP, (OR) 49.2, followed by SpA features, (OR) 11.3 and increased CRP, (OR) 10.3, AUC 91.4%, (p<0.001).

**Inter-observer agreement (IOA)** was good for BME. The mean (range) between the 1[st] reader (rheumatologist) and the 2[nd] reader (radiologist) was 0.70 (0.59; 0.81) (p <0.001). 3rd reader´s opinion (2[nd] radiologist) was employed for discordant cases, achieving 1.0 agreement. The IOA in our study was 0.76 (0.64; 0.88) for SPBME $\geq$2, and 0.59 (0.43; 0.75) for WPBME $\geq$2. Erosion $\geq$1 was assessed in 28.6% of the patients, according to the 1[st] reader (rheumatologist), in 19.5%, according to the 2[nd] reader (radiologist), and in 51.9%, according to the 3[rd] reader (radiologist). IOA between readers for Tw1CL was as follows; erosion 0.47 (0.35; 0.59), backfill 0.51 (0.35; 0.66), fat metaplasia 0.44 (0.32; 0.56), and sclerosis 0.53 (0.41; 0.64) (p<0.001) (see S5 Table).

## Discussion

Early SpA diagnosis remains challenging. Diagnosis based on conventional radiography has demonstrated a diagnostic delay by 7–10 years and the introduction of sacroiliac joints MRI has resulted in a significant improvement in the evaluation and the management of these patients. Semi coronal STIR has long been the commonly used sequence due to its ability to detect acute inflammatory changes. Recent incorporation of Tw1 sequence has shown to improve the diagnostic utility of the MRI method [17]. MRI-SIJ´s utility has been tested in several studies over the last years (see S5 Table). BME was shown as the most sensitive individual value by Jones, and the combination of BME and/or erosion was found to increase the sensitivity and the specificity of the tests. The values for the latter varied from 0.35 to 0.91 for sensitivity, and from 0.75 to 0.90 for specificity. According to the authors, that could be explained by the patient's cohort variety, the definition used for reference standard and the number of the evaluated MRI lesions [18]. Global evaluation (Tw1 and STIR) compared to ASAS definition (STIR) was assessed by Aydin, and a greater specificity was found for the global evaluation, 0.94 than for the ASAS definition, 0.89. In contrast, the ASAS definition was more sensitive. The study identified a cut-off value of BME $\geq$ 2 as the best combination of specificity and sensitivity [19]. Sensitivity 0.48 and specificity 0.92 of ASAS definition was reported by Joven, however, STIR sequences were performed in only one third of the cases [20]. In our study we tried to evaluate the diagnostic utility of BME $\geq$2 related to the level of confidence of the readers with the ASAS definition of highly suggestive BME and assess the reliability of the structural lesions, especially in patients with doubtful STIR imaging where BME was present but it was hard to determine whether the ASAS "highly suggestive" criterion was met. We found WPBME $\geq$2 more sensitive 0.87, but less specific 0.66, compared to the SPBME $\geq$2. The differences could be explained by the fact that SPBME $\geq$2 might correspond to late disease stages or higher disease activity, while patients with WPBME $\geq$2 could be a mixed group of early disease stages, lower disease activity or NSBP. Sensitivity and specificity reported in our study were similar to prior studies and we confirmed the fact that a cut-off value of BME $\geq$2 provided the best specificity without significantly lowered sensitivity. Unfortunately, there was a growing evidence that BME could be frequently reported in patients with NSBP and in healthy controls. Weber found a single BME lesion in 26.9% of NSBP patients and in a 22% of the healthy controls, while BME $\geq$ 2, meeting the ASAS criterion was observed in 23.1% of the NSBP patients and in 6.8% of the healthy controls [21]. Our study confirmed these data. We reported WPBME $\geq$ 2 in 33.7% of the NSBP patients, while SPBME $\geq$2 was seen in only 5.8% of them. This lack of specificity of the BME led to an attempt to find better definition, and focused attention on erosion. Weber, in the above study reported a great specificity for erosion, detected in only 3.8% of the patients with NSBP and in 1.7% of the healthy controls [21]. Thereafter, Maksymowych confirmed that chronic inflammatory lesions could be found in the SIJ of patients with nr-axSpA and highlighted the fact, that erosion might be detected even in the absence of MRI inflammation, most often in younger male HLA B27 positive patients [13]. Weber, again, showed a better specificity for erosion $\geq$1, as well as for the association of BME $\geq$2 with erosion $\geq$1, compared to the BME $\geq$ 2 alone. The study highlighted the fact that the cut-off value of affected SIJ necessary to achieve a predefined specificity of >0.9 corresponded to BME $\geq$2 and erosion $\geq$1 [22]. Candidate criteria based on BME $\geq$3 or 4 and erosion $\geq$2 was proposed, bringing enhanced sensitivity, 0.83 and specificity, 0.85 [23]. BME $\geq$3 was also reported to be highly specific in a study of asymptomatic patients with recurrent acute anterior uveitis, where erosion $\geq$2 was found to noticeably increase specificity, 0.96, although sensitivity significantly decreased, 0.47. In the same study, backfill showed the highest specificity range, 1, however, sensitivity was even poorer, 0.29 [24]. High specificity for Tw1CL was

likewise reported by Hooge, where the presence of at least five erosions or fatty lesions were seen to assured specificity of 0.95 [25]. Once again, the specificity of the Tw1CL was profoundly assessed in a Cross-sectional multicentre study including patients with recent onset axial SpA (DESIR cohort) and patients with NSBP (ILOS study). The study reported a high specificity for erosion ≥3, 0.90, fat metaplasia ≥3, 0.90 and, for the combination of at least five structural lesions, 0.9, nevertheless sensitivity was extremely poor, 0.30 [26]. Our study confirmed these data and we furthermore found that addition of erosion ≥1 to the ASAS criterion substantially improved specificity, especially in cases where STIR imaging was doubtful and it was hard to determine if the "highly suggestive criterion" was met. Despite the high specificity demonstrated by erosion over the last years, and recently mentioned by the ASAS working group as the lesion genuinely related to SpA, its usefulness was widely questioned because of its low inter-observer agreement. New techniques such as a volume-interpolated breath-hold examination sequence showed to be helpful to improve the erosion´s utility [27]. Future studies are necessary to confirm these data and clarify the roll of erosion in early SpA diagnosis.

The predictive values of backfill and fat metaplasia were less studied. Hu et al, reported backfill as a specific sign of SpA, frequently detected in patients with AS (78.8%), followed by nr-axSpA (11.1%). In patients without SpA, this percentage was only 1.8%, and lesions were related to infections or malignancies. No backfill was found in healthy controls. Specificity for the whole group was high, 0.98, nonetheless sensitivity was moderate 0.59 [28]. Our study confirmed these data. We found backfill sensitivity even lower, probably due to the fact that two-thirds our cases were represented by patients with nr-axSpA. Furthermore, we found that the addition of erosion and backfill to the ASAS criterion markedly improved specificity but did not contributed to early diagnosis. IOA in our study was good for BME, but fair for chronic lesions. Low agreement for Tw1CL was previously reported. Rueda showed fair agreement between local radiologist and expert radiologist, 0.45 for BME, and 0.31, for Tw1CL. Results improved when compared between rheumatologist and expert radiologist, 0.69 for BME alone, and 0.73 for BME and Tw1CL together [29]. IOA was further assessed in bone marrow, subchondral bone, ligaments, join capsule and space by Heuft- Dorenbosch. The study showed better agreement for BME in right/left SIJ, 0.73/0.65 respectively, than for chronic lesions, right/left SIJ, 0.37/0.66 respectively [30]. The prior use of SPARCC online training module has demonstrated, to improve substantially, the inter-reader reliability [31].

The predictive validity of the different ASAS items have been frequently assessed over the last years. The predictive values of ASAS definition was evaluated in a study of 708 patients with a recent (>3 months and<3 years) IBP from the DESIR cohort. The study confirmed the excellent PPV of the ASAS criterion as a whole, and of each of its "imaging" and "clinical" arms, 91%, 97% and 82% respectively [32]. We found positive HLA B 27 antigen in 49% of the SpA patients, specificity 80%, PPV, 57% and two third of our population were female. HLA B27 positivity varied from study to study and differences could be explained by the study design and the variety of patient's cohort. Positive HLA B 27 was found in 61.5% of the patients from the DESIR cohort and there were slightly more women (54%) than men. The study demonstrated that HLA B27 positivity was related to younger age at the onset of the IBP, less delay in diagnosis, lower frequency of psoriasis and higher frequency of MRI inflammation (spine and SIJ) and radiographic sacroiliitis [33]. The HLA B27 positivity and gender performances were also scored in three worldwide known cohorts of patients with recent IBP; ASAS, DESIR and SPACE. The study revealed male gender in 46% of the ASAS cohort, 47% of the DESIR cohort and 38% of the SPACE cohort. HLA B27 was positive in 52% of the ASAS cohort, (43% vs 9%), SpA/ not SpA patients; 58% of the DESIR cohort (32% vs 27%), SpA/ not SpA; and 43% of the SPACE cohort (36%vs7%) SpA/not SpA patients. The HLA B27 positivity was independently associated with an axial SpA diagnosis in each cohort [34]. Furthermore, HLA B 27

negativity in combination with negative MRI at the onset might predict negative MRI imaging during the follow up period [35]. Differences between men and women with recent-onset axial SpA were reported in a prospective multicentre French cohort, where women had higher disease activity despite having less radiographic sacroiliitis or MRI inflammation, more peripheral involvement and more family history, whereas male gender was frequently related to positive HLA-B27 antigen, elevated CRP, and MRI inflammation of the spine [36]. These differences have been seen to lead to a longer delay in diagnosis in women compared to men as shown in a cross-sectional study of Spanish population. The study revealed that 30.2% of the men received a first correct SpA diagnosis compared to only 11.1% of the women (p = 0.016) with more medical than patients delay. 77.1% of the men compared with 59.3% of the women were HLA B27 positive in the study (p = 0.02) [37]. Future studies are necessary to clarify these gender differences. Our study showed that IBP was a good marker for selecting patients with SpA, followed by SpA features and increased CRP. Similar data were reported by van Hoeven, where ASAS questionnaire for IBP, good response to NSAIDs, FH, and symptoms duration were the strongest predictors of axial SpA [38]. IBP defined by the combination of Calvin or ASAS criteria with alternating buttock pain in addition, was further found to markedly increase the possibility of a positive MRI image by Navarro-Compan [39].

## Treatment strategy

Despite the fact that the first anti-TNF alpha therapy trials showed a failure to prevent radiographic progression, today, there is strong evidence, that early treatment might change the disease course. Over the last years, several studies reported that new bone formation was more likely to be developed in advanced inflammatory lesions through a process of fat metaplasia. Syndesmophytes were found to be developed from type B corner inflammatory lesions, and that led to evidence that early inflammatory lesions might resolve without sequelae, while resolution of inflammation in the more advanced corner lesions with reparative changes, resulted in a new bone formation [40]. This turned into a new hypothesis of a window of opportunity and highlighted the fact that early treatment might influence the radiographic outcomes [41]. Limited spinal progression was observed in the Swiss Clinical Quality Management cohort during 2 years follow up, where prior use of TNF alpha blockers reduced the odds of progression by 50%, and in a placebo-controlled trial with Certolizumab after 4 years follow up, with less progression during the years 2–4 [42,43]. Promising results in reducing radiographic progression in a large proportion of patients (>80%) was found with the interleukin-17A blocking (Secukinumab) in 2 years follow up [44]. That sustained efficacy was confirmed in 4 years follow up, where no-radiographic progression was demonstrated in 79% of the patients [45]. Future studies are necessary to confirm those results and find better therapeutic strategies.

## Conclusions

At the onset, IBP might be a good marker for selecting patients with suspicion of SpA. The addition of erosion >1 to the ASAS criterion might be helpful for early diagnosis, especially in patients with doubtful STIR imaging where BME is present but it is hard to determine whether the ASAS "highly suggestive criterion" is met.

## Supporting information

**S1 Table. MRI parameters.**
(DOCX)

**S2 Table. Statistical analysis.**
(DOCX)

**S3 Table. Predictive values.**
(DOCX)

**S4 Table. Multivariate logistic regression binary analysis.**
(DOCX)

**S5 Table. Tw1 and STIR MRI imaging reliability.** Study comparison.
(DOCX)

## Acknowledgments

To my colleagues who participated in this study and to my Hospital Marina Salud, Denia, Alicante, España.

## Author Contributions

**Conceptualization:** Liliya Yankova Komsalova.

**Data curation:** Liliya Yankova Komsalova, Mª Pilar Martínez Salinas, José Fermín Gómez Jiménez.

**Formal analysis:** Liliya Yankova Komsalova.

**Funding acquisition:** Liliya Yankova Komsalova.

**Investigation:** Liliya Yankova Komsalova, Mª Pilar Martínez Salinas, José Fermín Gómez Jiménez.

**Methodology:** Liliya Yankova Komsalova, Mª Pilar Martínez Salinas, José Fermín Gómez Jiménez.

**Project administration:** Liliya Yankova Komsalova.

**Resources:** Liliya Yankova Komsalova.

**Software:** Liliya Yankova Komsalova.

**Supervision:** Liliya Yankova Komsalova.

**Validation:** Liliya Yankova Komsalova.

**Visualization:** Liliya Yankova Komsalova.

**Writing – original draft:** Liliya Yankova Komsalova, José Fermín Gómez Jiménez.

**Writing – review & editing:** Liliya Yankova Komsalova.

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
