## [Decision Letter · Decision Letter 0]

11 Jun 2020

PONE-D-19-35692

IMPACT OF EROSION ON THE EARLY DIAGNOSIS OF SPONDYLOARTHRITIS. A PROSPECTIVE 4 YEAR STUDY OF 133 PATIENTS.

PLOS ONE

Dear Dr. Komsalova,

Thank you for submitting your manuscript to PLOS ONE. After careful consideration, we feel that it has merit but does not fully meet PLOS ONE’s publication criteria as it currently stands. Therefore, we invite you to submit a revised version of the manuscript that addresses the points raised during the review process.

We look forward to receiving your revised manuscript.

Kind regards,

Natasha McDonald

Associate Editor

PLOS ONE

Journal Requirements:

2. In ethics statement in the manuscript and in the online submission form, please provide additional information about the patient records used in your retrospective study. Specifically, please ensure that you have discussed whether all data were fully anonymized before you accessed them and/or whether the IRB or ethics committee waived the requirement for informed consent. If patients provided informed written consent to have data from their medical records used in research, please include this information.

3. Your ethics statement must appear in the Methods section of your manuscript. If your ethics statement is written in any section besides the Methods, please move it to the Methods section and delete it from any other section. Please also ensure that your ethics statement is included in your manuscript, as the ethics section of your online submission will not be published alongside your manuscript.

"No specific funding was received from any bodies in the public,

commercial or not-for-profit sectors to carry out the work described in this article.

The main author will apply for grant from “Fundation Valenciana de Reumatología”,

address Avda de la Plata 34, 46013, Valencia, ES CIF G-53286977. Requirements for the

grant, final acceptance by high impact factor journal.".

" The funders had no role in study design, data collection and analysis, decision to publish, or preparation of the manuscript.".

Reviewers' comments:

Reviewer's Responses to Questions

**Comments to the Author**

1. Is the manuscript technically sound, and do the data support the conclusions?

Reviewer #1: Yes

Reviewer #2: Partly

2. Has the statistical analysis been performed appropriately and rigorously? 

Reviewer #1: I Don't Know

Reviewer #2: Yes

3. Have the authors made all data underlying the findings in their manuscript fully available?

Reviewer #1: Yes

Reviewer #2: Yes

4. Is the manuscript presented in an intelligible fashion and written in standard English?

Reviewer #1: Yes

Reviewer #2: Yes

5. Review Comments to the Author

Reviewer #1: I would like to congratulate the authors for their study on “Impact of erosion on the early diagnosis of SpA. Prospective 4 year study of 133 patients.” The article is well structured and articulated.

However, there are a few points which I would like to highlight -

1. The objective in the abstract section doesn’t correlate well with the study title. You May consider restructuring that sentence/ paragraph so that the aim of the study and the study title emerges to be having the same primary aim. For instance you can try something like - “To analyse predictive value of…. In early diagnosis of SpA”

2. In the methods section just cross check whether you meant to mention Ankylosing Spondylitis with abbreviation AE or AS?

Further in the methods section the authors mention that “All STIR images were blinded to patient’s data….. What about other sequences? Were all the sequences of each MRI were blinded or only the STIR sequences? Kindly clarify.

3. It would be nice if you provide appropriate headings/ title to all the tables for better clarity.

4. In table 2, it seems that the Not SpA and Yes SpA N, i.e. Number has not been mentioned along with the % in each. If it is 86 and 47 in each making up the total to 133, then please do mention that number in the appropriate place.

5. The discussion section is very well written, however for sake of completion and adding upto the discussion about the PPV and PNV of various criteria, I would recommend to go through the predictive values of various criterias and add the below article -

Evaluation of the Predictive Validity of the ASAS Axial Spondyloarthritis Criteria in the DESIR cohort by Meghnathi B et al. Clin Exp Rheumatol Sep-Oct 2019;37(5):797-802. 

6. Finally do cross check for a few typographical error

Reviewer #2: This study aims to assess the impact of MRI lesions in the sacroiliac joint on the early diagnosis of axSpA. It is concluded that the addition of erosion to BME detected by MRI enhances specificity for diagnosis without significantly compromising sensitivity. I have several suggestions to improve the quality of the manuscript.

1. There is no relevant introduction to this manuscript. The introduction should include a synopsis of the scientific question and/or unmet need followed by a statement of study hypothesis and/or objectives. This needs to be re-written.

2. Methods: “To define STIR acute inflammatory lesions (BME) we used the ASAS/ OMERACT definition, accepting for positive MRI the presence of BME ≥2, depicted as a strong hyperintense signal similar to that of the blood vessels”.

What does the score of BME≥2 mean and what is the origin of this score?

3. Methods: “We divided STIR sequences in three categories, strongly positive (SP) BME ≥2, if images fulfilled ASAS/OMERACT criterion and they were clearly presented and easily recognizable as positive by readers (high level of confidence), and weakly positive (WP) BME ≥2, if lesions fulfilled ASAS/OMERACT criterion, but they were tenuous and not easily identifiable as positive (low level of confidence).”

The rationale for using this method should be supported by a citation. Many would argue that this is a flawed method because previous work has shown that up to 40% of healthy individuals will have BME meeting the ASAS-OMERACT criterion.

4. Methods: “Sacroiliac radiographies were evaluated by the rheumatologist and assessed according to The Modified New York criteria (9). Radiographies were graded as positive if there were at least bilateral grade 2, or unilateral grade 3 sacroiliitis.”

The radiographs should have been assessed by 2 readers plus adjudicator just like the MRI scans.

5. The statistics section does not indicate what is the primary outcome, specifically how the final diagnosis was ascertained.

6. Results: “The average age in our study was 38.9 years with male/female ratio 0.41/1. 47 (35.3%) patients received diagnosis of SpA. 8 (17%) patients had radiographic sacroiliitis. IBP was found in 55 (41.4%) patients, positive HLA B 27 antigen in 40 (30.3%), increased CRP in 25 (18.8%) and SpAF in 25 (18.8%).”

SpA is typically a male disease with B27 prevalence >80%. In the SpA patients B27 positivity was only 50% casting doubt on the diagnostic process.

7. Results: “SP BME ≥2 (78.7% vs 6%), sensitivity, 0.79, specificity 0.94 (p<0.001), as well as, WP BME ≥2 (87% vs 33.7%), sensitivity 0.87, specificity, 0.66 (p<0.001).”

It should be stated how reliably SP versus WP was detected.

8. Results: “Those patients had also more often erosion ≥1 (72% vs 7%), (p<0.001).”

Which patients does this refer to,….the SP BME?

9. Results: “ROC curve, plotting true positive rate against false positive rate, with area under the curve (AUC) at least 50% was calculated, showing AUC, 91.6%, for patients with nr-axSpA, and 91.4%, for the total group (p<0.001).”

It should be clarified what variable is being analyzed in the AUC.

10. Results: “IOA between readers for T1wCL was as follows; erosion 0.47 (0.35; 0.59), backfill 0.51 (0.35; 0.66), fat metaplasia 0.44 (0.32; 0.56), and sclerosis 0.53 (0.41; 0.64) (p<0.001) (see table 5).”

This does not appear to be sufficiently reliable. It should be stated what training or calibration was undertaken by the readers.

6. PLOS authors have the option to publish the peer review history of their article (what does this mean?). If published, this will include your full peer review and any attached files.

Reviewer #1: No

Reviewer #2: No

---

## [Author Response · Author response to Decision Letter 0]

2 Nov 2020

Dear Editor,

We are incredibly grateful to you for your comments. Bearing in mind your suggestions we tried to improve the quality of our work and address your points of concern. 

We have now submitted as it was required the follows files: 

1) Cover letter

2) Rectified manuscript.

3) Rectified manuscript with track changes.

4) S1 Table 

5) S2 Table 

6) S3 Table 

7) S4 Table

8) S5 Table

9)esponse to the 1st reviewer.

10) Response to the 2nd reviewer.

Ethics statement has been moved to Methods section as it was suggested. 

Finding statement has been removed from Acknowledgments Section of our manuscript and we declare than the funders had no role in study design, data collection and analysis, decision to publish, or preparation of the manuscript.

---

## [Decision Letter · Decision Letter 1]

7 Dec 2020

PREDICTIVE VALUES OF INFLAMMATORY BACK PAIN, POSITIVE HLA B27 ANTIGEN AND ACUTE AND CHRONIC MAGNETIC RESONANCE CHANGES IN EARLY DIAGNOSIS OF SPONDYLOARTHRITIS. A STUDY OF 133 PATIENTS.

PONE-D-19-35692R1

Dear Dr. Komsalova,

We’re pleased to inform you that your manuscript has been judged scientifically suitable for publication and will be formally accepted for publication once it meets all outstanding technical requirements.

Kind regards,

Domokos Máthé

Academic Editor

PLOS ONE

Additional Editor Comments (optional):

In my view this amended manuscript does mirror the intended changes by the reviewers. It cold also serve as a clinical decision-help tool for clinicians eventually reading it. I proposed it be accepted for publication in PLoS ONE.

Reviewers' comments:

Reviewer's Responses to Questions

**Comments to the Author**

1. If the authors have adequately addressed your comments raised in a previous round of review and you feel that this manuscript is now acceptable for publication, you may indicate that here to bypass the “Comments to the Author” section, enter your conflict of interest statement in the “Confidential to Editor” section, and submit your "Accept" recommendation.

Reviewer #1: All comments have been addressed

Reviewer #2: (No Response)

2. Is the manuscript technically sound, and do the data support the conclusions?

Reviewer #1: Yes

Reviewer #2: Yes

3. Has the statistical analysis been performed appropriately and rigorously? 

Reviewer #1: I Don't Know

Reviewer #2: Yes

4. Have the authors made all data underlying the findings in their manuscript fully available?

Reviewer #1: Yes

Reviewer #2: Yes

5. Is the manuscript presented in an intelligible fashion and written in standard English?

Reviewer #1: Yes

Reviewer #2: Yes

6. Review Comments to the Author

Reviewer #1: Congratulations to the authors! The re-written manuscript has come out well. Well done.

Reviewer #2: The manuscript is greatly improved. The additional edits should primarily clarify that this data reflects an association study in a cross-sectional cohort and not a study analyzing the predictive capacity of MRI cut-offs in a longitudinal study. Only a single time point per patient was analyzed in this data set. The edits below are of a relatively minor nature.

1. Title: “PREDICTIVE VALUES OF INFLAMMATORY BACK PAIN…..”

The term ‘predictive values’ is not appropriate. “Association of …” would be more appropriate.

2. Abstract: “Objectives: To analyse the predictive values of inflammatory back pain…”

The term ‘predictive values is not appropriate. ‘Associations’ would be more appropriate.

“Factors forecasting positive diagnosis were IBP, followed by SpA features and

increased CRP.”

The term ‘forecasting’ is not appropriate. ‘Associated’ would be more appropriate.

3. Introduction: “First, to evaluate the predictive validity of….” and “to find factors forecasting positive…..”

The authors are not testing predictive validity or forecasting but testing associations.

4. Methods: “All Tw1 and STIR images were blinded for time order…..”

This is a bit confusing since there was only 1 time point for imaging for each patient.

5. Methods: “Multivariate logistic regression binary analysis was used to determine factors forecasting positive diagnosis”.

The term ‘forecasting’ is not appropriate and should be replaced with a term such as “association “.

6. Results: The first paragraph should state how many patients did not have a diagnosis of axSpA.

7. Results: The title “Predictive Values’ should be changed to something like “Variables associated with diagnosis of axSpA”.

8. Results: “Table 4 shows a multivariate binary analysis, searching for factors forecasting a positive diagnosis.”

The term ‘forecasting’ is not appropriate. “Associated with” would be better.

9. Tables:” The use of the term ‘predictive’ should also be corrected in the Tables.

7. PLOS authors have the option to publish the peer review history of their article (what does this mean?). If published, this will include your full peer review and any attached files.

Reviewer #1: No

Reviewer #2: No

---

## [Editor Report · Acceptance letter]

9 Dec 2020

PONE-D-19-35692R1 

Predictive values of inflammatory back pain, positive HLA B27 antigen and acute and chronic magnetic resonance changes in early diagnosis of Spondyloarthritis. A study of 133 patients. 

Dear Dr. Komsalova:

I'm pleased to inform you that your manuscript has been deemed suitable for publication in PLOS ONE. Congratulations! Your manuscript is now with our production department. 

Kind regards, 

on behalf of

Dr. Domokos Máthé 

Academic Editor

PLOS ONE